# Chemodiversity of Dissolved Organic Matter in the Amazon Basin

Michael Gonsior[1], Juliana Valle[2], Philippe Schmitt-Kopplin[3,4], Norbert Hertkorn[4], David Bastviken[5], Jenna Luek[1], Mourad Harir[4], Wanderley Bastos[6], and Alex Enrich-Prast[2,5]

[1]Chesapeake Biological Laboratory, University of Maryland Center for Environmental Science, Solomons, MD20688, USA
[2]Departamento de Ecologia, Universidade Federal do Rio de Janeiro, Rio de Janeiro, 21941-901, Brazil
[3]Analytical Food Chemistry, Technische Universität München, Freising-Weihenstephan, 85354, Germany
[4]Research Unit Analytical BioGeoChemistry, Helmholtz Zentrum Muenchen, Neuherberg, 85764, Germany
[5]Department of Thematic Studies - Environmental Change, Linköping University, Linköping, 581 83, Sweden
[6]Laboratory of Environmental Biogeochemistry, Universidade Federal do Rondônia, Rodovia, 76801-974, Brazil

*Correspondence to*: Michael Gonsior (gonsior@umces.edu)

**Abstract.** Regions in the Amazon Basin have been associated with specific biogeochemical processes, but a detailed chemical classification of the abundant and ubiquitous dissolved organic matter (DOM), beyond specific indicator compounds and bulk measurements, has not yet been established. We sampled water from different locations in the Negro, Madeira/Jamari and Tapajós River areas to characterize the molecular DOM composition and distribution. Ultrahigh resolution Fourier transform ion cyclotron resonance mass spectrometry (FT-ICR-MS) combined with excitation emission matrix (EEM) fluorescence spectroscopy and Parallel Factor Analysis (PARAFAC) revealed a large proportion of ubiquitous DOM but also unique area-specific molecular signatures. Unique to the DOM of the Rio Negro area was the large abundance of high molecular weight, diverse hydrogen-deficient and highly oxidized molecular ions deviating from known lignin or tannin compositions, indicating substantial oxidative processing of these ultimately plant-derived polyphenols indicative of these black waters. In contrast, unique signatures in the Madeira/Jamari area were defined by presumably labile sulfur and nitrogen-containing molecules in this white water river system. Waters from the Tapajós mainstem did not show any substantial unique molecular signatures relative to those present in the Rio Madeira and Rio Negro, which implied a lower organic molecular complexity in this clear water tributary, even after mixing with the mainstem of the Amazon River. Beside ubiquitous DOM at average H/C and O/C elemental ratios, a distinct and significant unique DOM pool prevailed in the black, white and clear water areas that were also highly correlated with EEM-PARAFAC components and define the frameworks for primary production and other aspects of aquatic life.

## 1 Introduction

With an average of about 200,000 m$^3$/sec and ~20% of the global freshwater discharge, the Amazon River transports larger amounts of riverine freshwater into the ocean than the next seven biggest river systems on Earth combined. DOM processed and transported within the Amazon Basin is therefore of global importance as 32-36 Tg of organic carbon (of which 60-70 % is dissolved) are exported seaward of Óbidos annually (1, 2). Repeated attempts have been made to describe the origin and

fate of particulate and dissolved organic matter throughout the Amazon River and its tributaries using a variety of methodologies. The sources of riverine OM have been identified through isotopic analysis of $\delta^{13}C$, $\delta^{18}O$, and $\delta^{15}N$ (3-8). Humic acids, fulvic acids, and lignin concentrations have also provided information on the OM sources (3, 4, 9) whereas labile DOM components such as saccharides and amino acids (10), as well as bacterial consumption (11) and biological and photo-degradation observations (12) have provided information on the OM reactivity and bioavailability. $^{13}C$ NMR and optical spectroscopy have been used to understand bulk characteristics and functional groups of Amazon organic matter (OM) (7, 13, 14) and OM age has also been analyzed using $\delta^{14}C$ measurements (15), but a detailed molecular understanding has not yet been established for large tributaries of the Amazon River Basin.

Amazon tributaries vary in their coloration and opacity due to their origin and reactivity and have traditionally been classified as "black waters", "white waters" and "clear waters" (16). These three water types play a continuing role in the transformation of OM, due to mediating light availability for aquatic life or photoreactivity. Black waters are influenced by chromophoric DOM (CDOM) and have low particulate mineral content (16, 17). It has been suggested that drainage areas of black water systems are characterized by moist, acidic, hydric soils that allow for leaching of terrestrially derived plant matter, like lignins, tannins and other plant materials that also contribute to the CDOM (17). White waters make up ~2/3 of the Amazon Basin (16, 18), exhibit low DOM levels and are less acidic (pH 6.6 ± 0.2) when compared to the black waters of the Rio Negro (pH 4.5 ± 0.9) (19). The "white" color reflects a high mineral particle load due to drainage and erosion of calcic sedimentary deposits (20) originating largely from the Andes mountains. Clear waters drain kaolinite clays (17) and contain high concentrations of iron and aluminum oxides that may adsorb humic acids (21). Because of removal of CDOM and suspended sediments prior to the water entering the mainstem of these clear water Amazon tributaries, the CDOM and suspended particle levels are typically very low in these systems. As a result, clear waterwaters are less light limiting and can support higher phytoplankton biomass, if the generally low nutrient levels are elevated (1, 16, 22). High precipitation in combination with low infiltration of flood plain soils produce rapid overland and shallow subsurface flow responses to precipitation events across the Amazon Basin which carry particulate organic matter (POM) and DOM to the river from the surrounding river corridor (6, 23).

High CDOM in black waters and suspended sediment concentrations in white waters limit light and therefore the autochthonous production of organic matter (22); accordingly, allochthonous inputs dominate the organic matter pool (9, 10, 15). In clear waters, light is abundant (22) but nutrients are limited and as a result, OM is still expected to be influenced by allochthonous input. However, agriculture and urbanization along clear waters can supply additional nutrients and therefore increase autochthonous OM production with potential consequences for the DOM pool. These specific physico-chemical properties of these three main types of waters in the Amazon Basin are expected to exhibit distinctly different organic matter signatures.

High bacteria counts (~109 L$^{-1}$) have been observed in black and white waters with peaks during annual floods (11) when terrestrial-derived DOM enters the river (8). The biodegradable DOM appeared to be rapidly removed from these waters causing bacterial carbon limitation (11). Clear waters showed phosphorus limitation during high water periods, but also

carbon limitation due to low carbon quality (24). Younger, biologically available organic matter is preferentially respired upstream and the apparent $\delta^{14}C$ age of POM increases downstream (11, 15). Therefore, a certain proportion of the DOM likely reflects the recalcitrant behavior after microbial metabolism.

The unique DOM environments found within the Amazon Basin, which are ultimately the drivers of aquatic life, have yet to be resolved on a fine scale. It is unclear how previous work on bulk DOM, optical properties, a few specific target compounds, isotopes, or microbial processes, reflect the authentic chemical diversity intrinsic to the complex OM found in black, white, and clear waters. Therefore, we investigated the chemodiversity of these three water types across the Amazon Basin by employing non-target FT-ICR-MS interfaced with soft ionization electrospray (ESI), which has enabled the characterization of thousands of individual molecular ions in complex DOM mixtures (25-33). By combining this technique with advanced optical characterization, excitation emission matrix (EEM) fluorescence spectroscopy and Parallel Factor Analysis (PARAFAC), we assess similarities and differences in DOM composition between different waters of the Amazon Basin.

Using these techniques, we attempted to answer a) what is the overall chemodiversity of DOM in the Amazon Basin, and do distinct differences in DOM composition exist between major Amazon tributaries and flooded area? b) How well do the simple optical properties represent the overall molecular composition of DOM as described by FT-ICR-MS in tropical ecosystems? Satisfactory comprehension of these relationships would have large implications for the understanding of aquatic food webs and also for predicting further transport and processing of DOM in the Amazon.

## 2 Materials and Methods

Water samples from the mainstem of the Madeira (and its small tributary Rio Jamari), Negro and Tapajós River were collected in duplicates using 1 L pre-combusted Pyrex glass bottles. The bottles were filled in the main stem of the river just below the surface. In addition, 9-10 lakes, that were flooded by the individual rivers at the time of sampling, were sampled in the same manner near the cities of Santarém (confluence of Tapajós and Amazon River: Rio Tapajós Area), São Carlos (Rio Madeira Area) and Novo Airão (Rio Negro Area) (Fig. S1). The sampling sites in the Rio Tapajós Area included samples from flooded clear water lakes adjacent to the mainstem of the Rio Tapajós and flooded lakes that were located after the confluence of Rio Tapajós and the mainstem of the Amazon River. However, the sampling sites after the confluence were still dominated by Rio Tapajós water. The collected samples were filtered and solid phase extracted either directly after collection (aboard river boats) or within three hours after collection, when smaller boats were used for sampling.

### 2.1 Solid-Phase Extraction

All 1 L water samples were filtered through pre-combusted (500 °C) Whatman GF/F glass fiber filters, acidified to pH 2 by using high purity formic acid (98%) and subsequently solid-phase extracted (SPE). The used SPE procedure has been previously described in detail (34), but it was modified and formic acid was used instead of HCl to prevent possible chloride

ion adduct formation in the ionization source of the FT-ICR-MS (35). Briefly, SPE-cartridges (Agilent, Bond Elut PPL, 1g resin), were activated with methanol (Chromasolv, Sigma Aldrich), rinsed with acidified ultrapure water (Milli-Q, pH 2, formic acid) and the acidified water samples were gravity-fed through the cartridges at a flow rate of ~10 mL min$^{-1}$. Subsequently, the cartridges were rinsed again with acidified Milli-Q water to remove remaining sample solution, dried and

eluted with 10 mL high purity methanol (Chromasolv, Sigma Aldrich) into pre-cleaned 40 mL amber glass vials. Methanolic samples were then kept on ice during the 2 weeks sampling period and later frozen at -18 °C. Methanolic extracts are stable when kept at -18 °C for extended periods of time (336). The adsorption efficiency of the used SPE method was in general 90-98 % of the chromophoric DOM (CDOM) (measured by EEM fluorescence) and 60-70 % of the dissolved organic carbon (DOC). All samples were run in duplicates. DOC concentrations for all samples ranged between 3-10 mg L-1, so only 1 L of

sample water was run through the SPE cartridge to prevent overloading the 1 g PPL resin.

**2.2 Nutrients, chlorophyll, DOC, total dissolved nitrogen (TDN) and total dissolved phosphorus (TDP) analyses.**

Small (40 mL) aliquots of each water sample were filter-sterilized (0.2 µm, Whatman GD/X Cellulose acetate filters) and stored on ice prior to analysis. Total dissolved nitrogen (TDN) and phosphorus analyses were performed by an automated FOSS® colorimetric flow injection analysis (FIA) system, according to the quality control guidelines recommended by the

manufacturer. The detection limit for TDN and total phosphorus was 5 µg L$^{-1}$ and standard deviations ranged between 0.0012 and 0.0023 mg L$^{-1}$ and between 0.0015 and 0.0021 mg L$^{-1}$, respectively. DOC was analyzed in triplicates by oxidation with sodium persulfate in a titanium oven under high temperature and high pressure by using an automatic carbon analyzer (InnovOx Sievers TOC Analyzer) with a detection limit of 0.05 mg L$^{-1}$ and standard deviation between 0.01 and 0.02 mg L$^{-1}$. Chlorophyll a was extracted and analyzed according to a previously published procedure (37) and analyzed

using a Turner Designs fluorometer (Model Trilogy) with a detection limit of 0.01 mg L$^{-1}$.

**2.3 Ultrahigh Resolution Mass Spectrometry**

All SPE samples were analyzed using negative mode electrospray ionization and a Bruker Solarix 12 Tesla FT-ICR-MS located at the Helmholtz Zentrum Munich, Germany. Details about the FT-ICR-MS analyses used in this study have been described previously (25, 30). Briefly, methanolic samples were diluted 1:20 with methanol and then directly inject into the

electrospray at a flow rate of 120 µL min$^{-1}$. Five hundred scans with a time domain of 4 megawords were averaged and the averaged spectra were post-calibrated using a list of known DOM internal calibrants. A mass accuracy of less than 0.2 ppm deviation from the actual mass was achieved (less than the mass of an electron). Multiple charged ions in flow injection FT-ICR-MS analysis may occur dependent on the solution concentration and electrospray condition parameters (38): however, in our analysis we only encountered singly charged ions as described previously (39). The achieved mass resolution was in

routine full scan 500,000 at *m/z* 400 and exact unambiguous molecular formulae were assigned to the observed molecular ions up to a *m/z* of 800 Da (40).

Negative mode electrospray ionization (ESI) typically generates several thousands of different *m/z* ions, but this ionization technique is largely biased toward organic acids, because of their high ionization efficiencies. For example, alcohols and saccharides do not ionize efficiently in electrospray and hence signals of these compound classes are largely lost in complex DOM mixtures analyzed by FT-ICR-MS. On the other hand, DOM contains a large diversity of highly polar and polyfunctional easily ionizable compounds (e.g. organic acids), operationally also classified as fulvic and humic acids.

Van Krevelen diagrams (41) were used to visualize the elemental ratios of unambiguously assigned molecular formulas. Kendrick plots (42) are also useful to determine members of homologous series, but we used a modified Kendrick plot, where the Kendrick mass defect (KMD) is divided by another independent parameter z* (33) to describe homologous series and molecular formulas only spaced by $CH_2$. This ratio of KMD divided by z* (KMD/z*) enabled the unambiguous determination of homologous series and an enhanced visualization (much better resolution between homologous series). Additional details about this approach were previously described (43).

## 2.3 Excitation Emission Matrix Fluorescence and Parallel Factor Analysis

CDOM was recovered almost quantitatively (>90 %) using the described SPE method and enabled a direct comparison of FT-ICR-MS results and optical properties of Amazon DOM. SPE-DOM samples were prepared for optical analyses using 100 µL of the methanolic extract that was completely dried under pure nitrogen; re-dissolved in 10 mL Milli-Q water and further diluted with Milli-Q water (1:20). The pH of the solutions of all re-dissolved samples ranged between 4-5 pH units. Optical properties are highly dependent on pH and hence a narrow pH range was beneficial to accurately compare samples. Excitation emission matrix (EEM) fluorescence measurements of dried and re-dissolved SPE-DOM were measured using a temperature-controlled Jobin Yvon Aqualog fluorescence spectrometer. The emission was recorded over the range from 200-600 nm for excitation wavelengths ranging from 240-600 nm at 3 nm intervals. Fluorescent intensities were Raleigh scattering corrected, inner filter corrected by using the absorbance data and normalized to a STARNA quinine sulfate fluorescent standard of 1 ppm concentration. The statistical Parallel Factor Analysis (PARAFAC) of fluorescence data (44) was applied in this study to the EEM data set by using drEEM, which was developed in Matlab and utilizes the N-way toolbox (45). Several models were tested and a 5 component model was developed on the normalized data (Fig. S2). This model showed the best results in terms of separation between components, residuals and core consistency. Normalization was reversed prior to split-half validation of 6 subsets of the 5 component model (45). The 5 component model was split-half validated and explained 99.86 % of the variability and the results of the split-half validation are given in supplementary information (Fig. S3). The maximum intensities of the components (Fmax) for each sample was exported and used for all subsequent statistical analyses.

## 2.4 Multivariate Statistical Analysis

Data mining and the application of multivariate statistics is increasingly important to be able to analyze very complex data sets. Examples of such multivariate approaches are hierarchical cluster analysis (HCA) and principle component analysis (PCA) that have been recently applied to ultrahigh resolution mass spectrometry (46-48) and EEM-PARAFAC (49). In this study, PCA and HCA were applied to mass spectrometry-based data sets of all collected spectra and their exact mass lists

and intensities. The duplicate samples were first averaged and resulted in up to 16,000 variables (*m/z* ions). A data matrix was compiled of all averaged samples and mass lists, where *m/z* ions were matched in a narrow 0.2 ppm error window. This matrix was then normalized by subtracting the average value from each data point and dividing it by the standard deviation (50). The resulting FT-ICR-MS mass list data sheet was then converted into a resemblance matrix by using Spearman Rank correlations and used to create hierarchical clusters. The same data set was also used in the PCA analysis of FT-ICR-MS

data. A similar approach was undertaken for the EEM-PARAFAC data set.

In an additional analysis, the two normalized data sets (FT-ICR-MS and EEM-PARAFAC) were combined to be able to determine hierarchical clusters on the variables (*m/z* ions and their intensities and EEM-PARAFAC Fmax values) and to create a microarray. This approach is in analogy to genetic data, where the expression levels of large numbers of genes can be simultaneously visualized. In our case, molecular formulas and their associated *m/z* ion intensities would be in analogy to

specific DNA sequences and concentrations. This approach enabled the generation of heat maps (software: TM4-Multi Experiment Viewer) and hierarchical clusters of all molecular formula assigned to *m/z* ions and also which *m/z* ions or molecular formula co-varied with EEM-PARAFAC Fmax values. This approach successfully depicted hierarchical clusters of specific classes of molecular formulas with distinct and confined chemodiversity as expressed in van Krevelen diagrams within the CHO, CHNO and CHOS pools (Fig. S7). Hence, subsets of molecular formulas that correlated well with EEM-

PARAFAC components were depicted.

## 3. Results and Discussion

Representative FT-ICR-MS spectra from each area are provided in Figure 1. The Rio Negro mass spectrum, in comparison to the other areas, clearly showed much higher intensities of hydrogen-deficient *m/z* ions in the low and high molecular weight ranges (Fig. 1). Further evaluation of these initial results by intensity-weighted parameters confirmed the trend from

higher mass to lower mass DOM molecules in the order Rio Negro, Rio Tapajós and Rio Madeira (Tab. 1). For example, the intensity-weighted center of mass with assigned CHO formulae was 453 Da in Rio Negro, 428 Da in Rio Tapajós and 419 Da in Rio Madeira. Counts of double bond equivalents (DBE) and DBE-oxygen (DBE-O) also followed the same decreasing pattern from Rio Negro, Rio Tapajós and Rio Madeira (Tab. 1). The intensity-weighted averaged oxygen to carbon (O/C) and hydrogen to carbon (H/C) elemental ratios did not follow the same decreasing trend and indicated that the displacement

of higher molecular weight DOM in the Rio Negro towards lower molecular weight in the Madeira or Tapajós Rivers was not associated with a change in overall elemental ratios. Somewhat similar trends where observed for the molecular ions that represented nitrogen-containing compounds, but the numbers of assigned CHNO formulae were approximately 30% lower in

the Rio Negro samples. Similarly, ions with sulfur-containing formulae were of very low abundance in the Rio Negro, when compared to higher occurrence in the Rio Tapajós and even higher proportions in the Rio Madeira (Tab. 1).

Detailed molecular formula assignments of duplicate SPE-DOM samples from the Rio Negro area (black water), Rio Tapajós area (clear water) and Rio Madeira area (white water) in both the main stem of the river as well as flooded adjacent lakes revealed an overall molecular composition of Amazon DOM that was extremely diverse (Fig. 2). CHO formulae covered almost the entire area of chemically reasonable O/C and H/C ratios, but also CHNO and CHOS formulae showed extensive compositional variance. The chemodiversity of all Amazon DOM signatures was characterized by 6,118 assigned molecular formulae occurring in duplicate samples of which 43 % were of hetero-atomic nature, i.e. CHNO, CHOS and CHNOS formulae (Fig. 2, top 6 panels). Common molecular formulae across all samples referred to 47 % of CHO, 31 % CHNO, but less than 2 % of CHOS molecular formulas (Fig. 2). However, distinct area-specific differences in relative abundances applied to many of the ubiquitous DOM $m/z$ ions (Fig.2, lower 6 panels).

A simple computation of unique signatures ($m/z$ ions) from each sampling area (Fig. 3) was combined with molecular formulae assignments and revealed that the Rio Negro area was characterized by unique, abundant and numerous high molecular weight compounds in a rather confined area within the van Krevelen diagram, indicative of a large diversity of polyphenolic-type compounds with high O/C (0.5-0.8) and low H/C (0.4-0.7) ratios, presumably highly degraded tannins or lignins. A large complement of the assigned formulae exceeded the common O/C ratios of known lignin or tannic acid sub-units and it appeared that these specific Rio Negro polyphenols were highly enriched in oxygen, possibly resulting from microbial side chain oxidation or even aromatic ring opening, which would lead to the incorporation of additional carboxylic acids (52).

A removal of these unique high molecular weight poly-aromatic DOM molecules by means of mineral adsorption or flocculation with iron (21) or aluminum oxides (53) after mixing with the high sediment-load Solimões River is conceivable and corresponded to previous reports that between 4% (54) and 40% (2) of the DOC was removed at the confluence of the Rio Negro and Solimões. We were not able to sample directly the Solimões River and hence a direct comparison between FT-ICR-MS results from the Rio Negro and Solimões were not achieved. However, investigations of Amazon wetland hydrogeochemistry also found analogous preferential sorption of higher molecular weight DOM to sediments, resulting in an enriched low molecular weight (LMW) aliphatic DOM pool under high suspended solids conditions (7).

In contrast, the distinct signatures indicative of the Rio Madeira area were comprised of diverse CHNO and CHNOS compounds that may define the readily bioavailable and labile DOM pool, because of in part its low molecular weight, in particular for the sulfur bearing $m/z$ ions. A specific CHOS pool of very low H/C ratios was also apparent in the Madeira area samples.. The unique and diverse CHNO and CHOS compounds observed in the Madeira area samples might have indicated anthropogenic influence as suggested in a recent study (54) and this suggestion was supported by the relatively high population closely to the Madeira sampling area. The city Porto Velho is about 30 miles upstream of the sampling locations and the whole area is intensively used for agriculture and to grow mainly soy beans. A potential source of sulfur are sulfonates which may origin from daily care products (e.g. surfactants), but also from wetting agents in fertilizers.

The Tapajós area showed indicative aliphatic CHO signatures (high H/C ratios) in the van Krevelen diagram and some diverse nitrogen-containing molecular formulae, but this region was in general characterized by ubiquitous molecular signatures found in all investigated areas.

EEM spectra of SPE-DOM also showed distinct differences between each area (Fig. S4), and a highly intense long-wavelength fluorescent peak in Rio Negro water samples was apparent (Fig. S4). A five PARAFAC components (Fmax1-5) model (Fig. S2) was most adequate to explain the differences in the fluorescence data set and also captured the indicative fluorescence signal of the Rio Negro (Fmax3 and 4) at high emission wavelengths (Fig. S5). These high emission wavelengths can either be explained by large complex conjugated π-systems (56) that would support the unique aromatic high molecular weight DOM characteristic of Rio Negro waters or charge transfer processes (57), which also would require rather complex molecular assemblies.

Additional multivariate analysis, such as HCA and PCA, of the FT-ICR-MS and EEM-PARAFAC data of all samples also produced distinct separation between sampling areas (Fig. 4). It was gratifying that the EEM-PARAFAC PCA and HCA clusters matched remarkably close to the results of the FT-ICR-MS data. These statistical results suggested that many of the molecular changes associated with each sampling area might be associated with certain changes in the fluorescent DOM (FDOM). This at least should apply for the Rio Negro area, because FT-ICR-MS-derived molecular signatures showed high hydrogen deficient molecules indicative of aromatic structures. However, correlations between optical properties (EEMs) and FT-ICR-MS data are not a certain proof of a causal relationship.

Traditionally, the classification of Amazon Rivers was based on appearance and color of the water, and largely defined by CDOM (e.g. black water - Rio Negro versus clear water - Rio Tapajós) and its sediment load (e.g. white water - Rio Madeira, Solimões). Hence, optical properties were important indicators at least for black and clear water systems. Our results suggested that this classification based on optical properties might expand to include specific molecular characteristics of waters from various Amazon Basin areas, but presumably non-fluorescent aliphatic CHOS and CHNO compounds were indicative for the Madeira sampling area and which were outside the analytical window of the EEM-PARAFAC approach (Fig. 4). Aromatic CHNO compounds, which were present in the Madeira area samples, presumably carry the ability to show long-wavelengths absorbance and fluorescence, even at relatively low molecular weight. At present, it remains unknown whether aromatic nitrogen heterocyclic compounds play an important role in FDOM.

Our results indicated that classified Amazon water systems (black, white and clear water) were associated with the presence or absence of many regionally unique compounds. It remains an open question what happens with the FDOM that is presumably adsorbed to the mineral phase or coagulated and become part of the particulate fraction and whether or not it is respired or transported to the Atlantic Ocean and eventually desorbed, or if it is added to the downstream sediments or to the seasonally flooded forest floor.

DOC concentrations in the Rio Negro (10.8 mg/L) and surrounding lakes (10.1 mg/L) were twice or more of that of Madeira and Tapajós area waters (1.9-5.8 mg/L) (Table 2), consistent with earlier observations with mean DOC concentrations in the Rio Negro of 12.7 mg/L, Madeira of 5.8 mg/L, Solimõesof 5.8 mg/L (not measured in this study) and the Tapajós of 4.5

mg/L (1, 2, 9). It was also previously stated that the Rio Negro does not produce a simple dilution effect at the confluence with the Solimões (2). The Solimões and in particular the Madeira River transport by far the highest load of suspended solids of all tributaries in the Amazon basin (2) and the suggested adsorption or more likely coagulation of DOC when mixed with high suspended sediment rivers is conceivable. However, a direct comparison between the Rio Negro and the Solimões at the confluence downstream of Manaus was not undertaken in this study.

High Rio Negro DOC concentrations may be indicative of watershed characteristics (17) or a lower mineral content (2, 7, 54) compared to the other tributaries. The DOC data were in agreement with the observations resulting from FT-ICR-MS (Fig.1 and 3) and EEM-PARAFAC analyses (Fig. S5) that indicated a removal of high molecular weight polyphenolic-like compounds from the Rio Negro waters, when compared to samples collected downstream or in other catchments. Total dissolved nitrogen (TDN) and total dissolved phosphorous (TDP) were low in all waters (Tab. 2), contributing to the observed low primary productivity despite the differences in light availability between the systems. A high DOM C:N ratio was observed in all waters consistent with previous findings (9, 58).

To address the question of which $m/z$ ions were correlated with each other and also with what specific EEM-PARAFAC Fmax components, both data sets were combined, normalized and correlated using Spearman Rank correlations (Fig. 5). All variables were statistically compared within the dissolved organic carbon (CHO), dissolved organic nitrogen (CHNO) and the dissolved organic sulfur (CHOS) pools (Fig. 5 and S6). Heat maps revealed clearly distinct hierarchical clusters in each of the CHO, CHNO and CHOS pools, indicative of a very similar behavior of variables within individual clusters. Remarkably, large clusters in the CHO pool were found which represented rather confined molecular ions with specific O/C and H/C ratios (Fig. S6). One unique CHO cluster consisted of hydrogen-deficient (low H/C), but highly oxygenated (high O/C) molecular ions and had a strong positive correlation with all Rio Negro samples (Fig. S7). Similarly, the Fmax components 3 and 4 were highly correlated to a cluster of high molecular weight molecular ions likely aromatic in origin (Fig. 5 and S7).. This unique DOM signature showed much higher O/C ratios when compared to known lignins and tannic acids and may resemble the result of humification and an increase in non-lignin aromatic structures and higher carboxyl group content (59).

In contrast, Fmax1,2 and 5 only correlated with a very few aliphatic molecular ions (Fig. 5 and S6), which were improbable fluorescent compounds itself, indicating an indirect correlation. Accordingly, correlations of Fmax 1, 2 and 5 were found within a distinct CHNO cluster, indicating that the fluorophores responsible for these PARAFAC components might have derived from heterocyclic or other aromatic nitrogen-containing molecules. However, components Fmax 3 and 4 did not show any correlation with molecular ions in the CHNO pool, nor in the CHOS pool, supporting the supposition that these PARAFAC components were only derived from CHO molecules.

The heat map and correlations of CHOS molecular ions (Fig. S6) reflected the already mentioned enrichment of CHOS compounds in the Madeira and Tapajós area samples and the absence of these signatures in the Rio Negro area. The unique highly hydrogen-deficient CHOS pool only found in the Madeira sampling area also manifested in a distinct cluster in this analysis. The CHOS-based cluster that co-varied with PARAFAC components Fmax1, 2 and 5 showed mostly molecular

ions with high O/C ratios, and a high probability to be aromatic indicated by a subset of formulae that showed H/C ratios below 1. In contrast to the CHO pool that correlated with Fmax 3 and 4, this specific CHOS pool was confined to the low molecular weight range. At this point, it appears that CHOS molecules also contributed to the fluorescence signals manifested in Fmax 1,2 and 5. Overall, Fmax 1, 2 and 5 always correlated together as did Fmax 3 and 4. A simple Spearman Rank correlation between the PARAFAC components confirmed the very strong correlations between these two clusters (Fig. S7).

Here, we present direct evidence that Rio Negro waters contained a unique, high molecular weight and highly fluorescent DOM component that is neither present in the Rio Madeira nor in the Rio Tapajós/Amazon confluence. This Rio Negro FDOM was confirmed by FT-ICR-MS to be indicative of high molecular weight polyphenolic-like compounds. DOC concentrations were also the highest in the Rio Negro. There is no reason to suspect widely different DOC produced by plants among the regions, but differences in geology is expected to influence the DOC pool, which is in part reflected in the large differences in pH and it is likely that this specific Rio Negro DOM fraction is sensitive to adsorption to mineral particles or coagulation with metals and that this specific DOM can be rapidly removed from the water during mixing of major Amazon tributaries as suggested in previous studies (2, 54). Such a removal would also substantially reduce the molecular complexity of Amazon DOM as observed in this study.

The similarities in DOM quality within regions between lakes and rivers (Tab. 2) indicated that the DOM composition is relatively stable under certain sets of environmental conditions. On the other hand, as discussed above, changing conditions may rapidly change DOM composition, in turn affecting conditions for aquatic life. While the commonly measured spectral properties of the DOM indicated these overall patterns, some important details were missed by solely measuring optical properties. Distinction between unique types of DOM with different origin, functions, and turnover times, such as the previously discussed CHNO and CHOS compounds, would not have been possible without FT-ICR-MS and multivariate statistical analyses. Thus, for improved capacity to understand and predict the fates and functions of DOM, new methods, such as ultrahigh resolution FT-ICR-MS, allowed detailed DOM characterization and emphasized the importance of non-target high resolution techniques.

## 4. Acknowledgements

This research was supported by the Brazilian research agencies FAPERJ, CAPES and CNPq and by the German research agencies Alexander von Humboldt and DAAD and the Swedish Research agencies STINT and VR. Enrich-Prast, A. has additional support as a research fellow from CNPq and a Cientista do Nosso Estado from FAPERJ. This is contribution XXXX (to be filled in after acceptance of manuscript) of the University of Maryland Center for Environmental Science, Chesapeake Biological Laboratory. All data used in this publication will be provided upon request by the corresponding author.

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

**Table 1: Differences in SPE-DOM between the Rio Negro, Rio Tapajós and Rio Madeira identified by ESI-FT-ICR-MS.**

| Intensity-weighted average values of all assigned molecular formulae (180-800 Da) | | | | | | | | |
|---|---|---|---|---|---|---|---|---|
| Formulae | sample | n[*] | Center of mass | $\Delta O:C_w$ | $\Delta H:C_w$ | $\Delta DBE_w$ | $\Delta DBE_w$-O | $\Delta DBE_w$:C |
| CHO | Negro | 3130 | 453 | 0.51 | 1.05 | 11.15 | 0.31 | 0.524 |
| | Tapajós | 3298 | 428 | 0.49 | 1.12 | 10.04 | 0.13 | 0.494 |
| | Madeira | 3151 | 419 | 0.51 | 1.10 | 9.93 | -0.03 | 0.504 |
| CHNO | Negro | 1362 | 429 | 0.49 | 0.99 | 11.58 | 1.85 | 0.590 |
| | Tapajós | 1975 | 413 | 0.49 | 1.02 | 10.92 | 1.68 | 0.580 |
| | Madeira | 2088 | 408 | 0.50 | 1.01 | 10.91 | 1.64 | 0.592 |
| CHOS | Negro | 58 | 313 | 0.22 | 1.84 | 2.49 | -0.99 | 0.143 |
| | Tapajós | 286 | 315 | 0.30 | 1.66 | 3.65 | -0.75 | 0.237 |
| | Madeira | 399 | 357 | 0.49 | 1.21 | 7.15 | -0.23 | 0.460 |

Note: $\Delta O:C_w$: Intensity-weighted averaged oxygen to carbon ratios of assigned molecular formulae; $\Delta H:C_w$: Intensity-weighted averaged hydrogen to carbon ratios of assigned molecular formulae; $\Delta DBE_w$: Intensity-weighted averaged double bond equivalency; $\Delta DBE_w$-O: $\Delta DBE_w$ with numbers of oxygen atoms subtracted, $\Delta DBE_w$:C: carbon normalized $\Delta DBE_w$. n: number of assigned molecular formulae to *m/z* molecular ions.

**Table 2: Averaged Nutrients, chlorophyll, DOC and TDN concentrations of all water samples collected in the main stem of the Rio Negro, Rio Madeira, Rio Jamari and Rio Tapajós as well as in 8-10 flooded lakes within the Rio Negro area, Rio Madeira area as well as within the Rio Tapajós and after its confluence with the Amazon River (Tapajós area) in May 2013.**

| | DOC$_{av}$ (mg/L) | DOC range (mg/L) | TDN (mg/L) | TDP (mg/L) | Chlorophyll a (μg/L) |
|---|---|---|---|---|---|
| Rio Negro area (10 flooded lakes and river) | 10.1 | 8.6-11.5 | 0.12 | 0.01 | 3 |
| Rio Tapajós area (9 flooded lakes and river) | 3.9 | 2.9-5.3 | 0.10 | 0.03 | 3 |
| Rio Madeira area (10 flooded lakes and river) | 2.9 | 1.9-5.8 | 0.11 | 0.03 | 2 |
| Rio Tapajós (mainstem river only) | 3.6 | | 0.13 | 0.05 | 2 |
| Rio Negro (mainstem river only) | 10.6 | | 0.12 | 0.02 | n.d |
| Rio Madeira (mainstem river only) | 5.8 | | 0.18 | 0.07 | n.d |
| Rio Jamari (mainstem river only) (white water tributary to the Madeira River) | 2.4 | | 0.23 | 0.04 | 2 |

5    Note: DOC$_{av}$: averaged DOC concentrations from each area. Standard error of DOC measurements were always between 0.1 and 0.2 mg -1. TDN: Total dissolved nitrogen; TDP: Total dissolved phosphorous. n.d. = not detected. Standard errors of TDN and TDP wer always between 0.001 and 0.0025 mg $^{-1}$

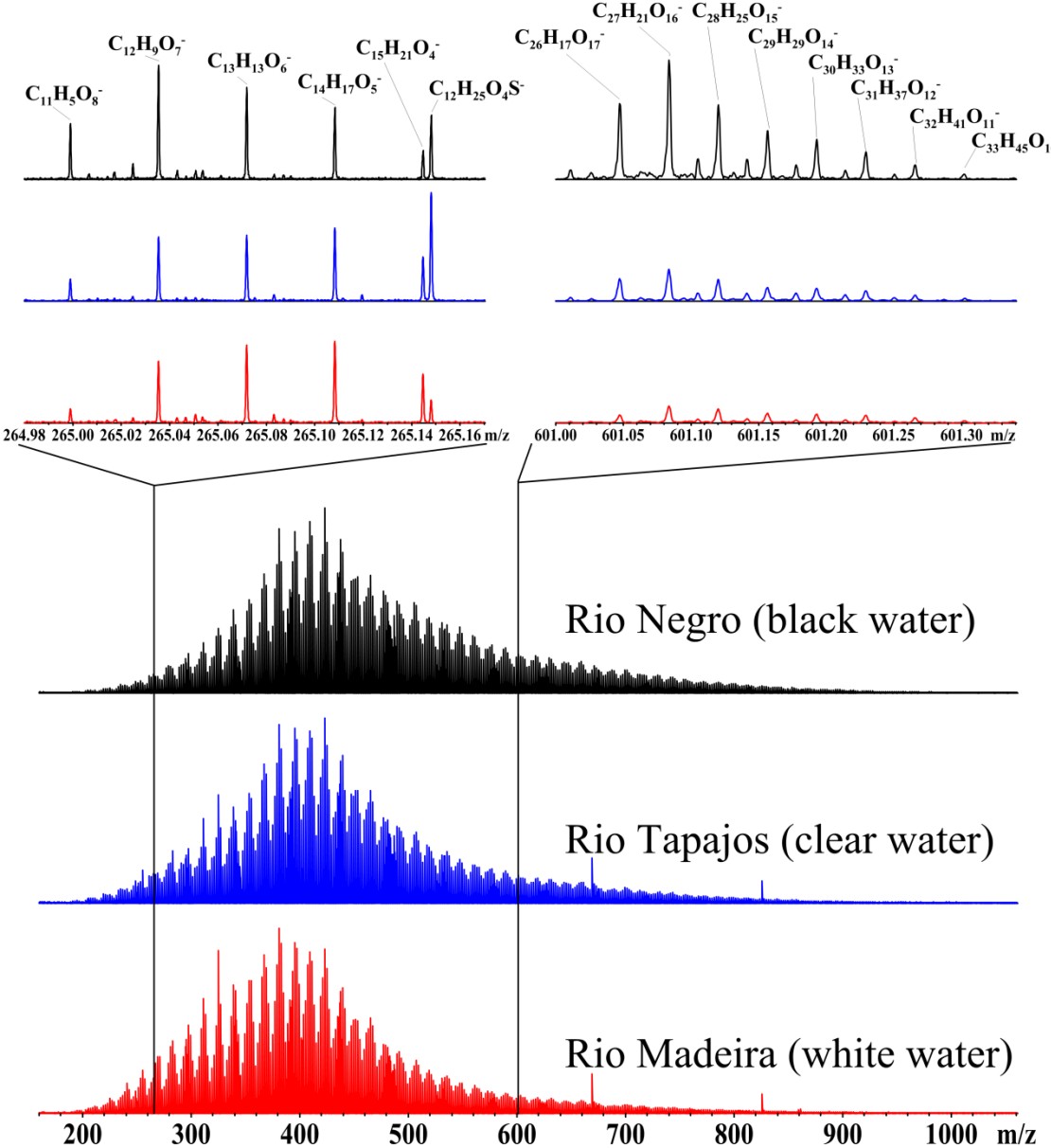

**Figure 2: Ultrahigh resolution mass spectra of SPE-DOM isolated from the mainstem of the Rio Negro, Rio Madeira and Rio Tapajós, Amazon, Brazil and the stacked relative abundances of all ions at nominal mass 265 and 601 for all three river systems (note the diferences in peak height for different colors, slightly phase shifted for increased visibility).**

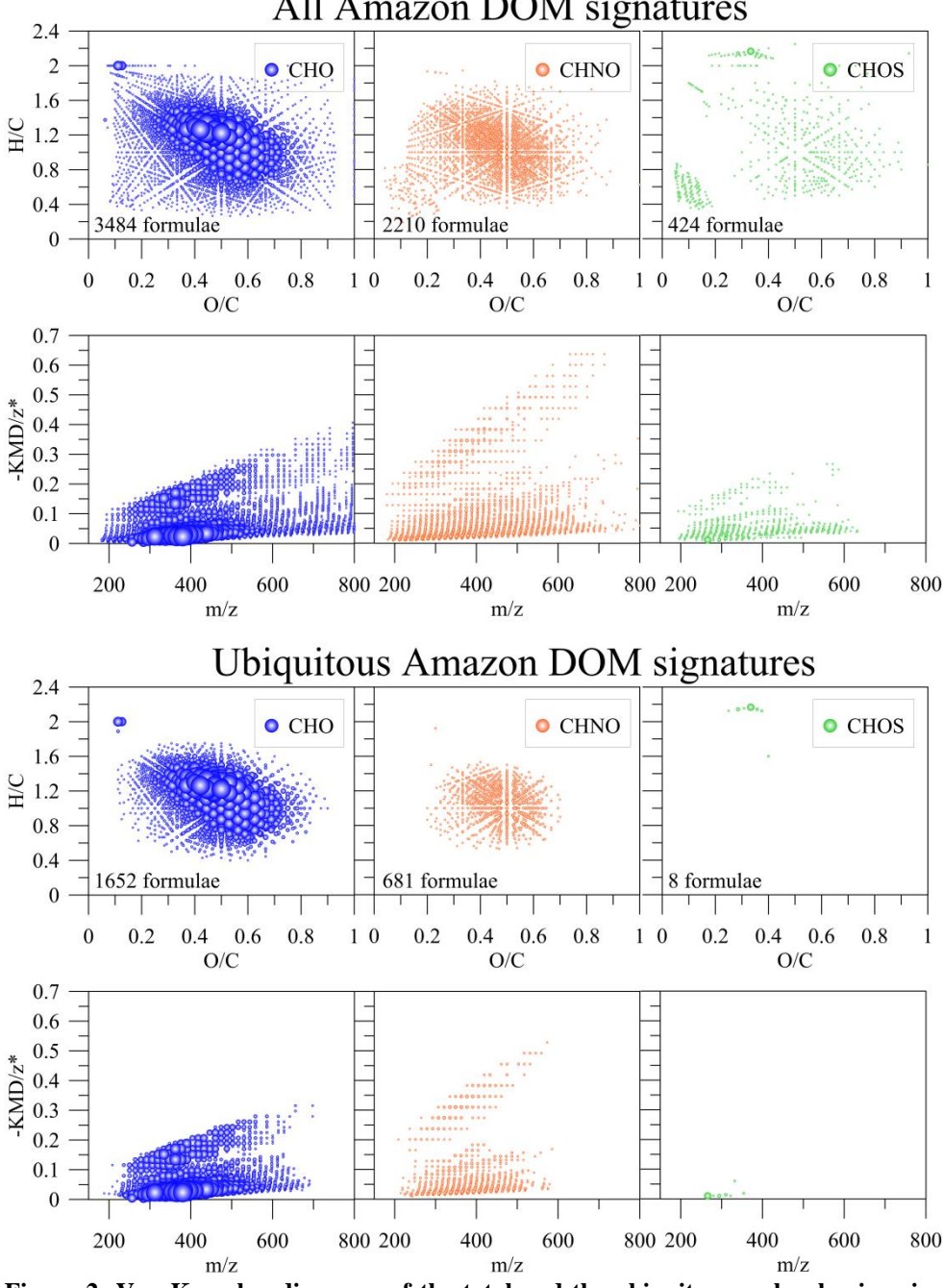

**Figure 2: Van Krevelen diagrams of the total and the ubiquitous molecular ions in all Amazon SPE-DOM samples analyzed by FT-ICR-MS.**

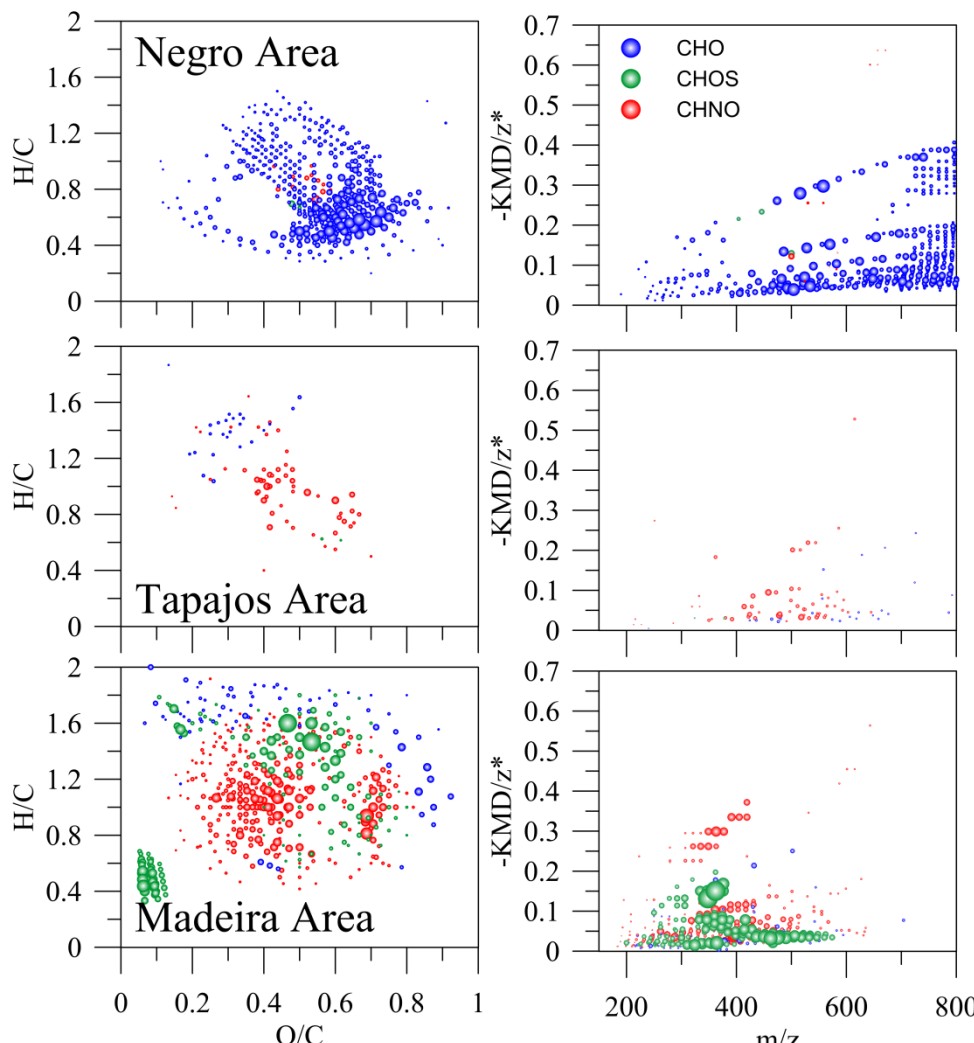

**Figure 3: Unique ions analyzed by FT-ICR-MS of solid-phase extracted DOM associated with the three different Amazon areas.**

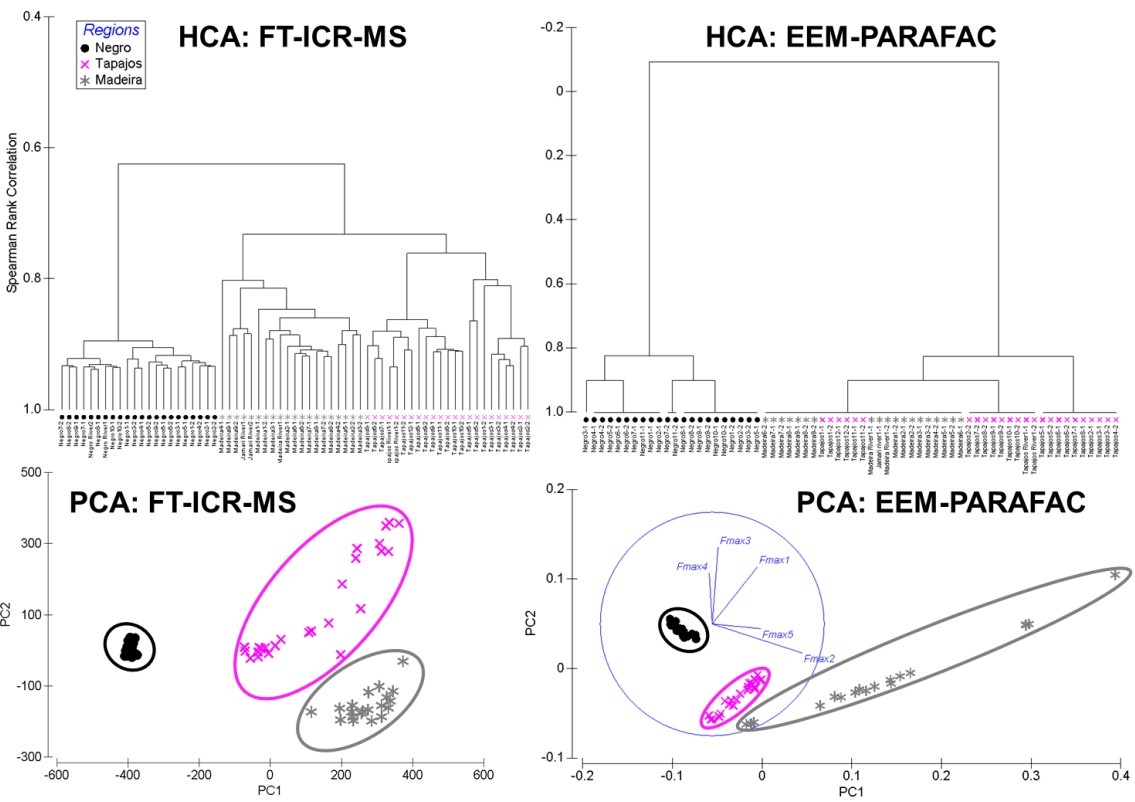

**Figure 4: Principal component analysis and hierarchical cluster analysis of the normalized data of all mass peaks and their intensities and all EEM-PARAFAC Fmax components.**

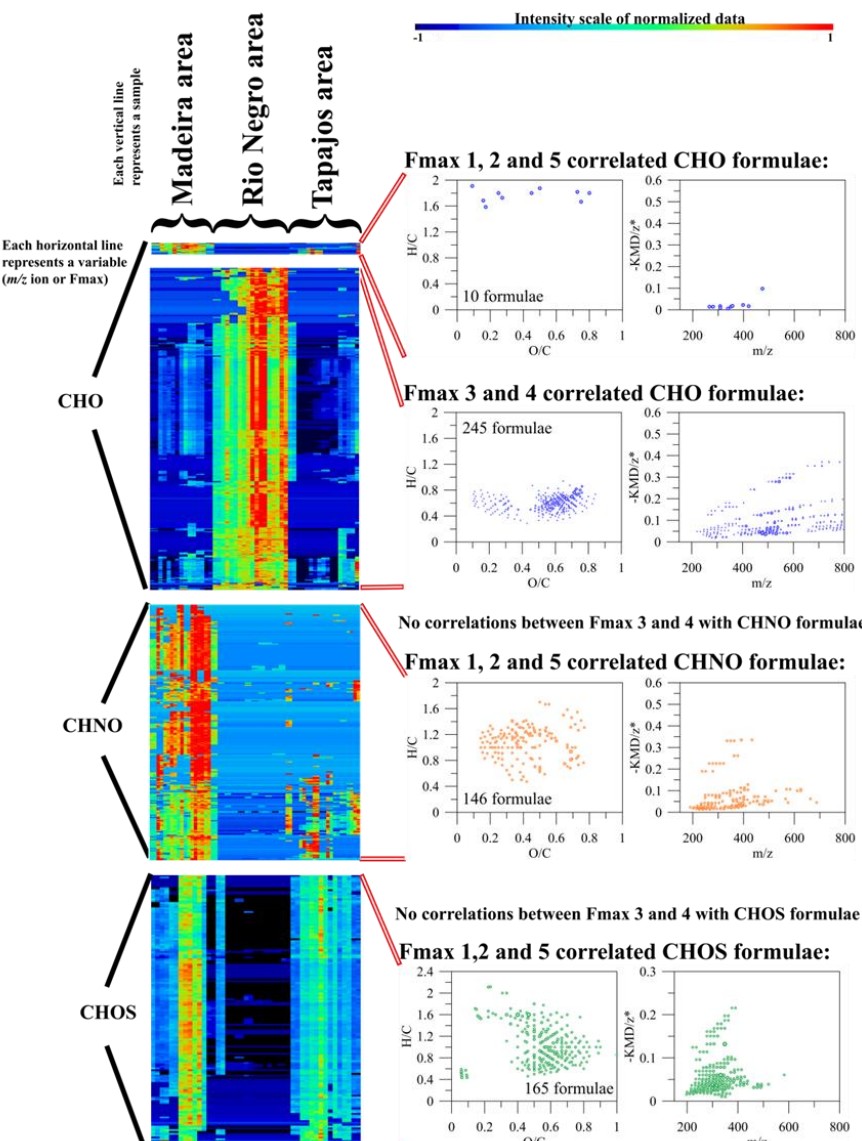

**Figure 5: Heat maps of specific Spearman Rank correlated hierarchical clusters (distance threshold 0.6) that showed correlations between *m/z* ions and Fmax values separated into the dissolved organic carbon (CHO) nitrogen (CHNO) and sulfur (CHOS) pool. The van Krevelen diagrams correspond to the molecular signatures that co-varied with specific EEM-PARAFAC Fmax values. The entire heat maps and additional specific *m/z* ion clusters are given in Figure S6, including their distribution in the chemical space visualized in van Krevelen diagrams.**