# Peer review of "Chemodiversity of Dissolved Organic Matter in the Amazon Basin"

_Biogeosciences, 2016_

## Referee Comment (RC1) · Anonymous Referee #1 · 22 Apr 2016

Review of Biogeosciences Discuss. Doi: 10.5194/bg-2016-72

This manuscript describes an interesting study combining EEMS and FT-ICR-MS analysis of SPE-extracted DOM samples from a black-water, a white-water, and a clear-water river within the Amazon basin. As such it provides one of the most detailed compositional studies of the DOM within a water system to date and provides key information to add to the large body of work on bulk organic matter characteristics within the Amazon Basin. The topic and general scientific approach make the study a good one for publication in this journal. The problem with the manuscript right now is that it appears to have been written assuming that the reader will know all the details of the analytical techniques and multivariate analyses. Key information is either missing or presented but not explained.

General comments: In Section 2.2., the limits of detection, and a measure of precision and accuracy need to be given for each technique. In section 2.3, the authors need to tell the reader whether positive or negative ionization is used and what solvent the samples were brought up in and what the mobile phase was, if there wasn't direct injection of the sample. The details promised in the supplementary file are not present. In Figure 1 (representative FT-ICR-MS spectra) and in Figure S2, the reader needs to know which samples these were, mainstem or some other location. Perhaps the stations in Figure S1 could be given individual names or numbers and these could be used in the captions to Figure 1 and S1? Three panels in Figure 3 have –KMD/z* as the y axis but the reader is never told what this means (I assume a Kendrick mass defect factor) nor are these panels discussed anywhere in the text or supplemental section. Finally, the description of the heat maps needs to be revised for clarity. I tried very hard through multiple readings of the text (with the figures printed out for quick reference) to understand exactly what was being correlated in each figure but was unable to do so. Could there be some guidance along the x and y axes and more description in the text of the technique itself as well as the results?

Specific comments:

1. The superscripts throughout this manuscript appear to have been lost in a formatting step.
2. Page 2, line 12, "has" should be "have"
3. P.3, line 11, "exists" should be "exist"
4. P. 4, line 7, should read "Small (40-mL) aliquots"
5. P. 4, line 18. Which convention should be used in this journal for five hundred thousand, the decimal or comma form?
6. P. 4, line 30. "One hundred microliters"?
7. Figure 1 is mislabeled in its caption as Figure 2.
8. P. 6, lines 3-6. This sentence is trying to say too many things and I cannot follow the points. Please reword. Perhaps start by pointing out the formulae for your hydrogen deficient vs more saturated compounds
9. P. 6, line 13, "where" should be "were"
10. P. 7, line 3, what does "disk-shaped" mean? You cannot know higher order folding of the molecules from MS data, can you? Or are you referring to a disk-shaped distribution of formulae in the van Krevelen?
11. P. 7, line 29, "close" should be "closely"
12. P. 7, line 31, "high" should be "highly"
13. P. 8, line 10, should read "long-wavelength absorbance"

14. P. 8, lines 8-13.  I am not sure I am following this reasoning.  Please clarify.
15. P. 8, lines 14-16. Please reword this sentence for better clarity.
16. P. 8, line 27, add "of"  before "which"
17. Please add information on the color scale and axis or variable labels for the heat maps.

---

## Referee Comment (RC2) · Anonymous Referee #2 · 27 Apr 2016

Anonymous Referee 2 Comments for Biogeosciences Discussions manuscript "Chemodiversity of dissolved organic matter in the Amazon Basin" MS number: bg-2016-72

Overall: The manuscript by Gonsior and coauthors compares the DOM quality using traditional optical methods (EEM, PARAFAC) with more specialized FT-ICR-MS and statistical methods in three different water types in the Amazon Basin. They present some interesting data and it is exciting that they saw definite relationships between the optical and molecular methods and the unique chemical signals between the three rivers. I think this paper would be improved by some rearrangement and additional content and clarification, especially regarding which samples are in the confluence of the Amazon River and how that relates to removal of DOM through adsorption and/or coagulation, currently the data presented does not clearly support DOM removal.

[Figure]

Specific Comments: For the most part, the sampling locations are divided into Rio Negro, Rio Madeira and Rio Tapajos. However, there are a couple of other location that are also presented e.g., Rio Jamari (abstract and tables), Solimoes River (discussion); additional information needs to be presented to give context to these locations or include the data under the three main location.

Are all of the samples from the Tapajos River in the confluence? It is not clear what stations are part of the confluence and which samples are just part of the river stem or flooded lakes. This needs to be clarified throughout the manuscript, but especially in the results and the discussion since a main point of your paper is the removal of DOM upon the mixing of rivers. In the current version of the manuscript the data is not presented to clearly demonstrate that DOM is being removed.

Pg 1, Line 21: Are all the samples from the Tapajos in the confluence? reword if this is not the case.

Pg 2, Line 19: Is it the removal of CDOM and suspended sediments that cause the waters to be clearer or the lack of CDOM and low suspended particle load due to clear waters draining weathered areas? Please clarify, add context, and/or estimates for each if available.

Pg 2, Lines 29-34: How do clear waters fit into this? Add information if available or add a statement to the effect that no information is available for clear waters.

Pg 2, Lines 29-34: Tie this paragraph into how it would relate to your study; link to changes in chemical compounds

Pg 3, Line 16: How many surface water samples were collected in the main stem of the river? Include range as you do for the lake samples. Specify how many and which samples were collected in the confluence vs the main stem.

Pg 3, Line 23: Add temperature and duration that filters were combusted.

Pg 4, Line 1: Samples were kept cool when? During extraction, until frozen? Please

clarify.

Pg 5, Line 1: What was the pH of the original river waters? If not pH 4-5, why was this pH chosen instead of a more neutral pH? Did you observe a pH dependence for your EEMs if samples were run at pH 4-5 vs pH 7-8?

Pg 7, Lines 3-7: The Solimoes River has not been mentioned before and no data are presented that show a decrease in DOM from the Rio Negro; the values for the flooded lakes and main river stem are almost identical for DOC. If the Solimoes River is just used as a comparison from another study, data needs to be presented that shows DOM removal occurred in the Rio Negro (DOC values and/or FT-ICR-MS figures from above and below the confluence).

Pg 7, Lines 9-19: This paragraph would benefit from some reorganization and clarification to allow this paragraph to flow better and increase understanding. Move the sentence "Very low abundance..." (lines 11-12) after the discussion about Rio Madeira and/or provide more context for the sentence, especially related to the boreal lakes and link to the next paragraph "The Tapajos areas contained..." (lines 17-19). Simplify the sentence "The unique and diverse..." (lines 13-15), are these the same or different unique compounds from the first sentence or the other unique compounds with unknown origin? Provide context as to how growing soy beans could cause unique compounds.

Pg 7, Line 23: How does Fmax3 relate to the Rio Negro? Fmax3 looks like it would also be a dominant component in Rio Negro in the example EEM in Fig. S2 and you state that Fmax 3 and 4 typically were correlated.

Pg 7-8, Lines 32-1: Unclear how the coagulation of analogous molecular ions fit into this paragraph. Suggest removing sentence.

Pg 8, Line 13-14: This sentence is vague and needs clarification. I agree that light attenuation would be different for the various sampling locations due to the differences

in the water characteristics, but how is this related to the unique compounds.

Pg 8, Lines 14-16: Additional discussion needs to be made to support this statement. No data is presented that suggests removal or transport of FDOM on particles or that you observed a loss of FDOM in the study area. If additional data are given to support the adsorption/coagulation, this statement could be expanded upon to tie into the bigger picture of biogeochemical cycling.

Pg 8, Lines 17-19: Please present evidence that the differences in the DOC concentration are a result of adsorption or coagulation besides just a difference in FT-ICR-MS. These three rivers seem to have very different geology in their headwaters that could also be responsible for the differences in DOC concentrations. This is especially relevant to the Rio Tapajos waters that should have a very low particle load.

Pg 8, Line 22: Could you provide some data to support the removal of HMW compounds from this river? Possibly a comparison of EEMs or FT-ICR-MS from upstream (lower particle load) and downstream (higher particle load)

Pg 9, Line 22-25: Additional information or clarification is needed for this section as stated previously regarding removal. The plants in the region may not vary substantially throughout the region but there is likely additional factors for the differences in DOC concentrations than just adsorption to mineral particles or coagulation with metals, likely a result of the geology in the different regions. Also no data is shown to support DOM removal through mixing.

Pg 16: Table 2: Rio Jamari is only mentioned in this table and the abstract. Provide some context for this location, is it another white water river like Rio Madeira? Or was this what the river was called after confluence with the Amazon River?

Pg 16: Table 2: It is not clear which data set is after the confluence with the Amazon River. Please provide clarification as Rio Tapajos has the same labeling (flooded lakes and main stem river) as the other rivers.

Pg 21, Fig. 5: To increase the understanding for readers unfamiliar with heat maps, include color bar legend and labels for the various axes.

Fig. S1: Are these stations for the flooded lake sites only or all sampling locations? Add text to caption for clarity. Can you provide additional information and make the map marker a different color for stations that were within the confluence with the Amazon River.

Technical Comments: Introduction:

Pg 2, Line 9-10: Switch "clear waters" and "white waters" since that is the order you talk about them.

Pg 2, Line 9: I suggest rewording the beginning of this paragraph to "Amazon tributaries vary in their coloration and opacity due to their origin and reactivity and have traditionally been classified as "black waters", "white waters" and "clear waters". These three water types play a continuing role..."

Pg 2, Line 11: Suggest using a different word than "processing" and simplifying "such as through"

Pg 2, Line 12: Change "has" to "have"

Pg 2, Line 23: Reword. Possibly to "High CDOM in black waters and suspended sediment concentrations in white waters limit light..."

Pg 3, Line 2-3: Reword. "It is unclear how previous work...or microbial processes, reflect the authentic..."

Pg 3, Line 10: Change "By applying" to "Using"

Pg 3, Line 10: Correct "overall"

Pg 3, Line 11: Simplify sentence by removing "moreover"

Pg 3, Line 13: Change "issues" to "relationships"

Materials and Methods: Pg 3, Line 16: Remove "by"

Pg 3, Line 17: Check number of lakes sampled, in Fig. S1, n=10,n=10, n=9 for Madeira, Negro, and Tapajos Rivers respectively. Should it be "9-10 lakes"? Or correct Figure S1.

Pg 3, Line 20: Move "either"; "were filtered and solid phase extracted either immediately after collection. . .or within three hours"

Pg 3, Lines 25-26: Reword sentence to "Formic acid was used instead of HCl to prevent possible chloride ion adduct formation. . .."

Pg 3, Line 28: Reword line 28 to include "pH 2, formic acid" from line 30, since this is the first occurrence of "acidified Milli-Q water"; if defined in line 28, do not need to include in line 30.

Pg 4, Line 4-5: Is this the range for all samples and location? Reword: "DOC concentrations for all samples ranged from 3-10 mg L-1, so only 1 L of sample water was filtered through the SPE cartridges to prevent overloading the 1 g cartridges."

Pg 4, Line 8-9: Move manufacture's name: ". . .performed by an automated FOSS$^{®}$ colorimetric flow injection analysis (FIA) system, according to the quality control guidelines recommended by the manufacturer"

Ultrahigh Resolution Mass Spectrometry Pg 4: I would recommend rearranging this paragraph so you introduce what you did before providing explanation.

Pg 4, Line 15-16: Did your instrument achieve a mass accuracy of less than 0.2 ppm? If so please clarify. State what your instrument did rather than saying what is typically achieved.

Pg 4, Line 25-26: Possibly move this sentence "All SPE samples. . ." to the beginning of the paragraph so the reader knows what you did prior to additional explanation.

Excitation Emission Matrix Fluorescence and Parallel Factor Analysis: Pg 4, Line 29-

30: Rearrange sentence: "CDOM was recovered almost quantitatively (>90

Pg 4, Line 30: Add volume to sentence "XX hundred uL..." or add a transition, "SPE-DOM samples were prepared by drying 100 uL of methanolic SPE-DOM..."

Pg 5, Line 12-13: This sentence could be simplified to "The maximum intensities of the components (Fmax) for each sample was exported and used for all subsequent statistical analyses."

Pg 5, Line 25: Which "two normalized data sets" are you referring to?

Results and Discussion: Pg 6, Line 2-6: Break into two sentences, "...and Rio Madeira. Conversely, at high m/z, e.g. NM 601..."

Pg 6, Line 10: Add "however" "However, the intensity-weighted..." Since the decreasing trend was not the same for elemental ratios as for the DBE.

Pg 6, Line 11: Replace 'that' with 'the' or 'the same', "did not follow the same decreasing trend..."

Pg 6, Line 14-16: Be more specific with how the different formulae were different in the various water types, e.g. instead of 'noticeably lower' maybe use 'approximately 50

Pg 6, Line 22: Clarify that these 6118 molecular formulae are for all the samples from all the waters combined and link to the top 6 panels of Fig. 2 "All Amazon DOM signatures"

Pg 6, Combine paragraph (lines 18-24) and next paragraph (25-27) or tie them together better.

Pg 6, Line 25: Remove "only"

Pg 6, Line 25-26: Tie this sentence to the lower 6 panels of Fig 2, possibly by using "ubiquitous" instead of "common".

Pg 7, Line 3: The phrase "disk-shaped" is odd, is there a better word choice.

Pg 7, Line 19: Add 'found', "signatures found in all"

Pg 7, Line 24: Provide reference for conjugated $\pi$-systems.

Pg 7, Line 29: add "to" "remarkably close to the results"

Pg 8, Line 7: Change 'However' to 'For example' or reword. This sentence seems like it is in support of the previous discussion that the optical properties could be related to specific molecular characteristics.

Pg 8, Lines 21-22: Simplify sentence: "The DOC data were in agreement with the FT-ICR-MS and EEM-PARAFAC results..."

Pg 8-9, Line 32-2: Simplify and reword. Possibly "One unique CHO cluster consisted of hydrogen-deficient (low H/C), but highly oxygenated (high O/C) molecular ions and had a strong positive correlation with all Rio Negro samples (Fig. S6). Similarly, the Fmax components 3 and 4 were highly correlated to a cluster of high molecular weight molecular ions likely aromatic in origin (Fig. 5 and S6)." Tie this back to lignin.

Pg 9, Lines 6-9: Simplify the sentence "Components Fmax 3 and 4..." (lines 6-9) to "However, components Fmax 3 and 4 did not show any correlation with molecular ions in the CHNO pool, nor in the CHOS pool, supporting the supposition that these PARAFAC components were only derived from CHO molecules." May fit better immediately after the rest of the discussion about Fmax 3 and 4, end of line 2.

Pg 9, Line 11: Remove comma and 'was'.

Pg 9, Line 16: Reword: "Overall, Fmax 1, 2, and always correlated together as did Fmax 3 and 4."

Pg 9, Line 21: Change 'appearance' to 'compounds'.

Pg 9, Line 27: Reference Table 2 and highlight the similarities between the river and flooded lake.

Pg 9, Line 30-33: Split into two sentences.

Pg 9, Line 34: Be specific about which methods. ". . .without FT-ICR-MS and statistical analyses"

Tables: Pg 16: Table 2: Give number of samples collected for each location.

Figures: Pg 17: Should be Figure 1

Supplemental Figures: Rearrange supplemental figures so that they are numbered as they are listed in the text.

Fig. S1: Rearrange the station order since you always talk about them in the order Rio Negro, Madeira, and Tapajos.

Fig. S1: Since this is a supplementary figure, possible to make maps larger?

Fig. S6: To increase the understanding for readers unfamiliar with heat maps, include color bar legend and labels for the various axes.

———————————————

---

## Author Comment (AC1) · 11 May 2016

**Reviewer 1**
Review of Biogeosciences Discuss. Doi: 10.5194/bg-2016-72

This manuscript describes an interesting study combining EEMS and FT-ICR-MS analysis of SPE-extracted DOM samples from a black-water, a white-water, and a clear-water river within the Amazon basin. As such it provides one of the most detailed compositional studies of the DOM within a water system to date and provides key information to add to the large body of work on bulk organic matter characteristics within the Amazon Basin. The topic and general scientific approach make the study a good one for publication in this journal. The problem with the manuscript right now is that it appears to have been written assuming that the reader will know all the details of the analytical techniques and multivariate analyses. Key information is either missing or presented but not explained.
General comments: In Section 2.2., the limits of detection, and a measure of precision and accuracy need to be given for each technique. In section 2.3, the authors need to tell the reader whether positive or negative ionization is used and what solvent the samples were brought up in and what the mobile phase was, if there wasn't direct injection of the sample. The details promised in the supplementary file are not present. In Figure 1 (representative FT-ICR-MS spectra) and in Figure S2, the reader needs to know which samples these were, mainstem or some other location. Perhaps the stations in Figure S1 could be given individual names or numbers and these could be used in the captions to Figure 1 and S1? Three panels in Figure 3 have –KMD/z* as the y axis but the reader is never told what this means (I assume a Kendrick mass defect factor) nor are these panels discussed anywhere in the text or supplemental section. Finally, the description of the heat maps needs to be revised for clarity. I tried very hard through multiple readings of the text (with the figures printed out for quick reference) to understand exactly what was being correlated in each figure but was unable to do so. Could there be some guidance along the x and y axes and more description in the text of the technique itself as well as the results?

**Limit of detection and the standard deviations for each method was added in section 2.2**
**Additional information about the FT-ICR-MS technique was added:** *"All SPE samples were analyzed using negative mode electrospray ionization and a Bruker Solarix 12 Tesla FT-ICR-MS located at the Helmholtz Zentrum Munich, Germany. Details about the FT-ICR-MS analyses used in this study have been described previously (25, 30). Briefly, methanolic samples were diluted 1:20 with methanol and then directly inject into the electrospray at a flow rate of 120 μL min$^{-1}$. Five hundred scans with a time domain of 4 megawords were averaged and the averaged spectra were post-calibrated using a list of known DOM internal calibrants."*
**Additional information were added to Figure 1. Sample FT-ICR-MS spectra were from the mainstem of the rivers.**
**Additional information about the visualization tools are given in the method section:** *"Van Krevelen diagrams (41) were used to visualize the elemental ratios of unambiguously assigned molecular formulas. Kendrick plots (42) are also useful to determine members of homologous series, but we used a modified Kendrick plot, where the Kendrick mass defect (KMD) is divided by another independent parameter z* (33) to describe homologous series and molecular formulas only spaced by CH2. This ratio of KMD divided by z* (KMD/z*) enabled the unambiguous determination of homologous series and an enhanced visualization (much better resolution between homologous series). Additional details about this approach were previously described (43)."*

**Additional explanations are now added to the method section and Figure 5 and S6 were revised and the color legend was added, as well as what the horizontal and vertical lines are.**

Specific comments:

1. The superscripts throughout this manuscript appear to have been lost in a formatting step.

**Corrected**

2. Page 2, line 12, "has" should be "have"

**Corrected**

3. P.3, line 11, "exists" should be "exist"

**Corrected**

4. P. 4, line 7, should read "Small (40-mL) aliquots"

**Corrected**

5. P. 4, line 18. Which convention should be used in this journal for five hundred thousand, the decimal or comma form?

**Corrected to use comma as separator**

6. P. 4, line 30. "One hundred microliters"?

**Corrected**

7. Figure 1 is mislabeled in its caption as Figure 2.

**Corrected**

8. P. 6, lines 3-6. This sentence is trying to say too many things and I cannot follow the points. Please reword. Perhaps start by pointing out the formulae for your hydrogen deficient vs more saturated compounds

**The sentence was simplified and it read now as follow: "*The Rio Negro mass spectrum, in comparison to the other areas, clearly showed much higher intensities of hydrogen-deficient m/z ions in the low and high molecular weight ranges (Fig. 1).*"**

9. P. 6, line 13, "where" should be "were"

**Corrected**

10. P. 7, line 3, what does "disk-shaped" mean? You cannot know higher order folding of the molecules from MS data, can you? Or are you referring to a disk-shaped distribution of formulae in the van Krevelen?

**The word disk-shaped was deleted. What was meant here is that these aromatics are planar and hence may undergo charge transfer if they get in close proximity.**

11. P. 7, line 29, "close" should be "closely"

**Corrected**

12. P. 7, line 31, "high" should be "highly"

**Corrected**

13. P. 8, line 10, should read "long-wavelength absorbance" \

**Corrected**

14. P. 8, lines 8-13. I am not sure I am following this reasoning. Please clarify.

**This statement was simplified to: "*…areas, but presumably non-fluorescent aliphatic CHOS and CHNO compounds were indicative for the Madeira sampling area and which were outside the analytical window of the EEM-PARAFAC approach (Fig. 4).*"**

15. P. 8, lines 14-16. Please reword this sentence for better clarity.

**The sentence was changed to: "*Our results indicated that classified Amazon water systems (black, white and clear water) were associated with the presence or absence of many regionally unique compounds. It remains an open……*"**

16. P. 8, line 27, add "of" before "which"

**Corrected**

17. Please add information on the color scale and axis or variable labels for the heat maps.

**The figures containing heatmaps were updated to include color legend and more details about horizontal (m/z values ot Fmax values) and vertical (samples) lines.**

**Reviewer 2**
Anonymous Referee 2 Comments for Biogeosciences Discussions manuscript "Chemodiversity of dissolved organic matter in the Amazon Basin" MS number: bg- 2016-72

Overall: The manuscript by Gonsior and coauthors compares the DOM quality using traditional optical methods (EEM, PARAFAC) with more specialized FT-ICR-MS and statistical methods in three different water types in the Amazon Basin. They present some interesting data and it is exciting that they saw definite relationships between the optical and molecular methods and the unique chemical signals between the three rivers. I think this paper would be improved by some rearrangement and additional content and clarification, especially regarding which samples are in the confluence of the Amazon River and how that relates to removal of DOM through adsorption and/or coagulation, currently the data presented does not clearly support DOM removal.

Authors Response: It was not possible during the sampling to directly compare Rio Negro waters with Solemoes waters at the confluence east f Manaus, but we added results from previous studies that strongly support a removal of DOM from the Rio Negro River. Our data supports these earlier findings and gives the first molecular fingerprint of this DOM component that is presumably adsorbed or coagulated. It was not the aim of this study to understand in detail the mechanism of this removal but we felt that a reasonable explanation would be coagulation with suspended solids transported by the Solemoes and Madeira Rivers.

Specific Comments: For the most part, the sampling locations are divided into Rio Negro, Rio Madeira and Rio Tapajos. However, there are a couple of other location that are also presented e.g., Rio Jamari (abstract and tables), Solimoes River (discussion); additional information needs to be presented to give context to these locations or include the data under the three main location. Are all of the samples from the Tapajos River in the confluence? It is not clear what stations are part of the confluence and which samples are just part of the river stem or flooded lakes. This needs to be clarified throughout the manuscript, but especially in the results and the discussion since a main point of your paper is the removal of DOM upon the mixing of rivers. In the current version of the manuscript the data is not presented to clearly demonstrate that DOM is being removed.

Authors Response: Additional information was added to the method section to explain in more detail where the samples were collected. The removal of Rio Negro DOM after mixing with the Solemoes has been demonstrated before and it is only one part of this study to show which DOM component is unique for the Rio Negro and hence is likely the one removed after mixing with high suspended sediment rivers. We tried to make this point clearer throughout the manuscript and please see responses to specific comments below.

Pg 1, Line 21: Are all the samples from the Tapajos in the confluence? reword if this is not the case.

No. Some samples were taking directly from the Tapajos and adjacent flooded lakes. (3). All other samples were collected in the confluence but most showed a much higher contribution from the Tapajos when compared to the mainstem of the Amazon. Hence, the area was named "Tapajos Area". More details were added in the method section to clarify this in more detail.

Pg 2, Line 19: Is it the removal of CDOM and suspended sediments that cause the waters to be clearer or the lack of CDOM and low suspended particle load due to clear waters draining weathered areas? Please clarify, add context, and/or estimates for each if available.

The lack of CDOM and suspended solids is the reason for the clear appearance of the Tapajos. We clarified this in the text.

Pg 2, Lines 29-34: How do clear waters fit into this? Add information if available or add a statement to the effect that no information is available for clear waters.

We added a reference to explain the Tapajos clear water system in more detail.

Pg 2, Lines 29-34: Tie this paragraph into how it would relate to your study; link to changes in chemical compounds

**We tried to improve this paragrapha and added a sentence:  …**"These specific physico-chemical properties of these three main types of waters in the Amazon Basin are expected to exhibit distinctly different organic matter signatures."

Pg 3, Line 16: How many surface water samples were collected in the main stem of the river? Include range as you do for the lake samples. Specify how many and which samples were collected in the confluence vs the main stem.

**Additional information about sampling locations was added to the method section**

Pg 3, Line 23: Add temperature and duration that filters were combusted.

**Added combustion temperature to method section**

Pg 4, Line 1: Samples were kept cool when? During extraction, until frozen? Pleaseclarify.

**Samples were kept on ice during the sampling period and later frozen. Details were added to the method section**

Pg 5, Line 1: What was the pH of the original river waters? If not pH 4-5, why was this pH chosen instead of a more neutral pH? Did you observe a pH dependence for your EEMs if samples were run at pH 4-5 vs pH 7-8?

**The pH was consistent for all samples after dissolving the dried SPE-DOM in Milli-Q water. A narrow pH range of all samples is important to not introduce a pH bias as shown in our previous publication (Timko, S. A., M. Gonsior, and W. J. Cooper. 2015. Influence of pH on fluorescent dissolved organic matter photo-degradation. Water Research 85: 266-274.)**

Pg 7, Lines 3-7: The Solimoes River has not been mentioned before and no data are presented that show a decrease in DOM from the Rio Negro; the values for the flooded lakes and main river stem are almost identical for DOC. If the Solimoes River is just used as a comparison from another study, data needs to be presented that shows DOM removal occurred in the Rio Negro (DOC values and/or FT-ICR-MS figures from above and below the confluence).

**Additional information from previous studies was added to make this point clearer. The Solimoes was unfortunately not sampled during our sampling campaign, but previous data strongly supports the removal of DOC from the Rio Negro after mixing with the Solemoes.**

Pg 7, Lines 9-19: This paragraph would benefit from some reorganization and clarification to allow this paragraph to flow better and increase understanding. Move the sentence "Very low abundance: : :" (lines 11-12) after the discussion about Rio Madeira and/or provide more context for the sentence, especially related to the boreal lakes and link to the next paragraph "The Tapajos areas contained: : :" (lines 17-19). Simplify the sentence "The unique and diverse: : :" (lines 13-15), are these the same or different unique compounds from the first sentence or the other unique compounds with unknown origin? Provide context as to how growing soy beans could cause unique compounds.

**The sentence of the comparison with boreal lake DOM was deleted, because it distracted from the findings. The paragraph was revised and a sentence was added at the end of the paragraph:**

**"**A potential source of sulfur are sulfonates which may origin from daily care products (e.g. surfactants), but also from wetting agents in fertilizers.**"**

Pg 7, Line 23: How does Fmax3 relate to the Rio Negro? Fmax3 looks like it would also be a dominant component in Rio Negro in the example EEM in Fig. S2 and you state that Fmax 3 and 4 typically were correlated.

**You are absolutely right. Fmax3 behaves like Fmax4. This was corrected in the text.**

Pg 7-8, Lines 32-1: Unclear how the coagulation of analogous molecular ions fit into this paragraph. Suggest removing sentence.

**The sentence was removed**

Pg 8, Line 13-14: This sentence is vague and needs clarification. I agree that light attenuation would be different for the various sampling locations due to the differencesin the water characteristics, but how is this related to the unique compounds.

**The sentence was clarified to: "*Our results indicated that classified Amazon water systems (black, white and clear water) were associated with the presence or absence of many regionally unique compounds.*"**

Pg 8, Lines 14-16: Additional discussion needs to be made to support this statement. No data is presented that suggests removal or transport of FDOM on particles or that you observed a loss of FDOM in the study area. If additional data are given to support the adsorption/coagulation, this statement could be expanded upon to tie into the bigger picture of biogeochemical cycling.

**Previously published literature values were added and an additional figure (S5) was added to the supplementary material to emphasize the much higher long-wavelengths EEM-PARAFAC components in the Rio Negro samples when compared to all other samples. The text was expended accordingly.**

Pg 8, Lines 17-19: Please present evidence that the differences in the DOC concentration are a result of adsorption or coagulation besides just a difference in FT-ICR-MS. These three rivers seem to have very different geology in their headwaters that could also be responsible for the differences in DOC concentrations. This is especially relevant to the Rio Tapajos waters that should have a very low particle load.

**Additional convincing literature data were added to the manuscript.**

Pg 8, Line 22: Could you provide some data to support the removal of HMW compounds from this river? Possibly a comparison of EEMs or FT-ICR-MS from upstream (lower particle load) and downstream (higher particle load)

**The text was clarified and links to the figures that already showed this in the FT-ICR-MS data are given. An additional Figure S5 was added to show the much enhanced intensity of long-wavelength absorbance and fluorescence in the Rio Negro samples.**

Pg 9, Line 22-25: Additional information or clarification is needed for this section as stated previously regarding removal. The plants in the region may not vary substantially throughout the region but there is likely additional factors for the differences in DOC concentrations than just adsorption to mineral particles or coagulation with metals, likely a result of the geology in the different regions. Also no data is shown to support DOM removal through mixing.

**Additional explanations were added to the manuscript**

Pg 16: Table 2: Rio Jamari is only mentioned in this table and the abstract. Provide some context for this location, is it another white water river like Rio Madeira? Or was this what the river was called after confluence with the Amazon River?

**The Jamari is a small high sediment load tributary to the Madeira but it is also flooded by the Madeira during the high water season. Some more details are given in the method section.**

Pg 16: Table 2: It is not clear which data set is after the confluence with the Amazon River. Please provide clarification as Rio Tapajos has the same labeling (flooded lakes and main stem river) as the other rivers. Pg 21, Fig. 5: To increase the understanding for readers unfamiliar with heat maps, include color bar legend and labels for the various axes.

**The table was clarified**

Fig. S1: Are these stations for the flooded lake sites only or all sampling locations? Add text to caption for clarity. Can you provide additional information and make the map marker a different color for stations that were within the confluence with the Amazon River.
**More explanation was added to the method section to make this clearer**

**Technical Comments: Introduction:**

Pg 2, Line 9-10: Switch "clear waters" and "white waters" since that is the order you talk about them.
**Corrected**
Pg 2, Line 9: I suggest rewording the beginning of this paragraph to "Amazon tributaries vary in their coloration and opacity due to their origin and reactivity and have traditionally been classified as "black waters", "white waters" and "clear waters". These three water types play a continuing role: : :"
**Corrected**
Pg 2, Line 11: Suggest using a different word than "processing" and simplifying "such as through"
**Changed wording and replaced "processing" to "transformation"**
Pg 2, Line 12: Change "has" to "have"
**Corrected**
Pg 2, Line 23: Reword. Possibly to "High CDOM in black waters and suspended sediment concentrations in white waters limit light: : :"
**Corrected**
Pg 3, Line 2-3: Reword. "It is unclear how previous work: : :or microbial processes, reflect the authentic: : :"
**Corrected**
Pg 3, Line 10: Change "By applying" to "Using"
**Corrected**
Pg 3, Line 10: Correct "overall"
**Corrected**
Pg 3, Line 11: Simplify sentence by removing "moreover"
**Corrected**
Pg 3, Line 13: Change "issues" to "relationships" Materials and Methods: Pg 3, Line 16: Remove "by"
**Corrected**
Pg 3, Line 17: Check number of lakes sampled, in Fig. S1, n=10,n=10, n=9 for Madeira, Negro, and Tapajos Rivers respectively. Should it be "9-10 lakes"? Or correct Figure S1.
**Corrected to 9-10 lakes**
Pg 3, Line 20: Move "either"; "were filtered and solid phase extracted either immediately after collection: : :or within three hours"
**Corrected**
Pg 3, Lines 25-26: Reword sentence to "Formic acid was used instead of HCl to prevent possible chloride ion adduct formation: : :."
**Corrected**
Pg 3, Line 28: Reword line 28 to include "pH 2, formic acid" from line 30, since this is the first occurrence of "acidified Milli-Q water"; if defined in line 28, do not need to include in line 30.
**Corrected**
Pg 4, Line 4-5: Is this the range for all samples and location? Reword: "DOC concentrations for all samples ranged from 3-10 mg L-1, so only 1 L of sample water was filtered through the SPE cartridges to prevent overloading the 1 g cartridges."

**Corrected**

Pg 4, Line 8-9: Move manufacture's name: ": : :performed by an automated FOSS® colorimetric flow injection analysis (FIA) system, according to the quality control guidelines recommended by the manufacturer" Ultrahigh Resolution Mass Spectrometry

**Corrected**

Pg 4: I would recommend rearranging this paragraph so you introduce what you did before providing explanation.

**The paragraph was changed as suggested.**

Pg 4, Line 15-16: Did your instrument achieve a mass accuracy of less than 0.2 ppm? If so please clarify. State what your instrument did rather than saying what is typically achieved.

**Yes it did. This was corrected.**

Pg 4, Line 25-26: Possibly move this sentence "All SPE samples: : :" to the beginning of the paragraph so the reader knows what you did prior to additional explanation. Excitation Emission Matrix Fluorescence and Parallel Factor Analysis:

**Moved this sentence to the beginning**

Pg 4, Line 29-30: Rearrange sentence: "CDOM was recovered almost quantitatively (>90

**Corrected**

Pg 4, Line 30: Add volume to sentence "XX hundred uL: : :" or add a transition, "SPEDOM samples were prepared by drying 100 uL of methanolic SPE-DOM: : :"

**Corrected**

Pg 5, Line 12-13: This sentence could be simplified to "The maximum intensities of the components (Fmax) for each sample was exported and used for all subsequent statistical analyses."

**Corrected**

Pg 5, Line 25: Which "two normalized data sets" are you referring to?

**More details were added.**

**Results and Discussion:**

Pg 6, Line 2-6: Break into two sentences, ": : :and Rio Madeira. Conversely, at high m/z, e.g. NM 601: : :"

**Corrected**

Pg 6, Line 10: Add "however" "However, the intensity-weighted: : :" Since the decreasing trend was not the same for elemental ratios as for the DBE.

**The trend was the same, but literature citations may have been confused with data. This was corrected!**

Pg 6, Line 11: Replace 'that' with 'the' or 'the same', "did not follow the same decreasing trend: : :"

**Corrected**

Pg 6, Line 14-16: Be more specific with how the different formulae were different in the various water types, e.g. instead of 'noticeably lower' maybe use 'approximately 50

**More specific details were added**

Pg 6, Line 22: Clarify that these 6118 molecular formulae are for all the samples from all the waters combined and link to the top 6 panels of Fig. 2 "All Amazon DOM signatures"

**This was clarified in the text.**

Pg 6, Combine paragraph (lines 18-24) and next paragraph (25-27) or tie them together better.

**Corrected**

Pg 6, Line 25: Remove "only"

**Corrected**

Pg 6, Line 25-26: Tie this sentence to the lower 6 panels of Fig 2, possibly by using "ubiquitous" instead of "common".

**Corrected**

Pg 7, Line 3: The phrase "disk-shaped" is odd, is there a better word choice.

**This was deleted for clarity.**

Pg 7, Line 19: Add 'found', "signatures found in all" Pg 7, Line 24: Provide reference for conjugated _-systems.

**Corrected**

Pg 7, Line 29: add "to" "remarkably close to the results"

**Corrected**

Pg 8, Line 7: Change 'However' to 'For example' or reword. This sentence seems like it is in support of the previous discussion that the optical properties could be related to specific molecular characteristics.

**This sentence was changed to: "…waters from various Amazon Basin areas, but presumably non-fluorescent aliphatic CHOS and CHNO compounds were indicative for the Madeira sampling area which were outside the analytical window of the EEM-PARAFAC approach (Fig. 4)."**

Pg 8, Lines 21-22: Simplify sentence: "The DOC data were in agreement with the FT-ICR-MS and EEM-PARAFAC results: : :"

**Corrected**

Pg 8-9, Line 32-2: Simplify and reword. Possibly "One unique CHO cluster consisted of hydrogen-deficient (low H/C), but highly oxygenated (high O/C) molecular ions and had a strong positive correlation with all Rio Negro samples (Fig. S6). Similarly, the Fmax components 3 and 4 were highly correlated to a cluster of high molecular weight molecular ions likely aromatic in origin (Fig. 5 and S6)." Tie this back to lignin.

**Paragraph was restructured and tied to a possible humification: "*One unique CHO cluster consisted of hydrogen-deficient (low H/C), but highly oxygenated (high O/C) molecular ions and had a strong positive correlation with all Rio Negro samples (Fig. S7). Similarly, the Fmax components 3 and 4 were highly correlated to a cluster of high molecular weight molecular ions likely aromatic in origin (Fig. 5 and S7). This unique DOM signature showed much higher O/C ratios when compared to known lignins and tannic acids and may resemble the result of humification and an increase in non-lignin aromatic structures and higher carboxyl group content (59).*"**

Pg 9, Lines 6-9: Simplify the sentence "Components Fmax 3 and 4: : :" (lines 6-9) to "However, components Fmax 3 and 4 did not show any correlation with molecular ions in the CHNO pool, nor in the CHOS pool, supporting the supposition that these PARAFAC components were only derived from CHO molecules." May fit better immediately after the rest of the discussion about Fmax 3 and 4, end of line 2.

**Moved and corrected**

Pg 9, Line 11: Remove comma and 'was'.

**Corrected**

Pg 9, Line 16: Reword: "Overall, Fmax 1, 2, and always correlated together as did Fmax 3 and 4."

**Corrected**

Pg 9, Line 21: Change 'appearance' to 'compounds'.

**Corrected**

Pg 9, Line 27: Reference Table 2 and highlight the similarities between the river and flooded lake.

**Referenced**

Pg 9, Line 30-33: Split into two sentences. Pg 9, Line 34: Be specific about which methods. ": : :without FT-ICR-MS and statistical analyses"
**Corrected**
Tables: Pg 16: Table 2: Give number of samples collected for each location.
**Added**
Figures: Pg 17: Should be Figure 1
**Corrected**
Supplemental Figures: Rearrange supplemental figures so that they are numbered as they are listed in the text.
**Corrected**
Fig. S1: Rearrange the station order since you always talk about them in the order Rio Negro, Madeira, and Tapajos.
**Corrected**
Fig. S1: Since this is a supplementary figure, possible to make maps larger?
**Maps were enlarged**
Fig. S6: To increase the understanding for readers unfamiliar with heat maps, include color bar legend and labels for the various axes.
Legend added